# The Challenge of Digital Transition in Engineering. A Solution Made from a European Collaborative Network of Remote Laboratories Based on Renewable Energies Technology

Macarena Martínez *, Francisca Segura and José Manuel Andújar

Research Centre CITES, University of Huelva, 21007 Huelva, Spain
* Correspondence: macarena.martinez@diesia.uhu.es; Tel.: +0034-959-21-73-80

**Abstract:** Society currently faces two crucial challenges: digital transition and energy transition. Educative innovation plays a key role in this challenging scenario, particularly engineering careers, where laboratory practices are as important as theoretical classes. This paper presents a standardized training platform supported by five European universities which include a remote laboratory experience. Each university is responsible for developing a training module under the guidance provided by the responsible entity (University of Huelva, Spain). For this purpose, the University of Huelva has implemented a remote laboratory based on a supercapacitor power bank. The rest of the universities have selected any other renewable source and have replicated the information and communications technology (ICT) infrastructure. The result is a European network materialized on a homogenized platform where teachers and students can find all the teaching materials (theory and practice) to train and to be trained in renewable energy matters in the new digital era.

**Keywords:** digital transition; engineering training; renewable energy education; remote laboratory; European university network

## 1. Introduction

The evolution of teaching has changed from the times when the teacher was a figure of considerable authority, and classes resembled monologues, to today, where education makes the student the protagonist of his or her own learning process, encouraging creativity and participation.

Apart from the teacher–student role and academic methodologies, since 2020, the face-to-face nature of the classroom has undergone significant changes. With the arrival of COVID-19 [1], the face-to-face nature of the classroom, especially higher education, has undergone significant changes. Modern higher education combines face-to-face classes with online learning [2].

This change in the method of teaching affects higher education [3–5], especially in scientific–technological areas, where a large part of the training depends on laboratory practices. Digital transition in education [6], especially in engineering, cannot be immediate, because students are unable to complete their training without practice with real systems. In order to adapt to this new digital scenario, remote laboratories appear as an interesting solution [7,8]. The main four types of laboratories are explained below and shown in Figure 1:

- Virtual laboratory with local access (Figure 1a): The environment is virtual and is accessed locally. All experimental work is performed in a computer simulation. The software tool is installed on a computer that meets the requirements to run it.
- Virtual laboratory with remote access (Figure 1b, virtual laboratory): The environment is simulated, and the student accesses through the Internet. The student uses an experimentation interface of a simulated system via the Internet.

- Real laboratory with local access (Figure 1c): This represents the face-to-face traditional laboratory where the student performs experiments in a physical plant located in the same room.
- Real laboratory with remote access (Figure 1d, remote laboratory): There is a real environment which the student accesses through the Internet. The user is able to manipulate the real plant from anywhere, thus providing greater facilities to students.

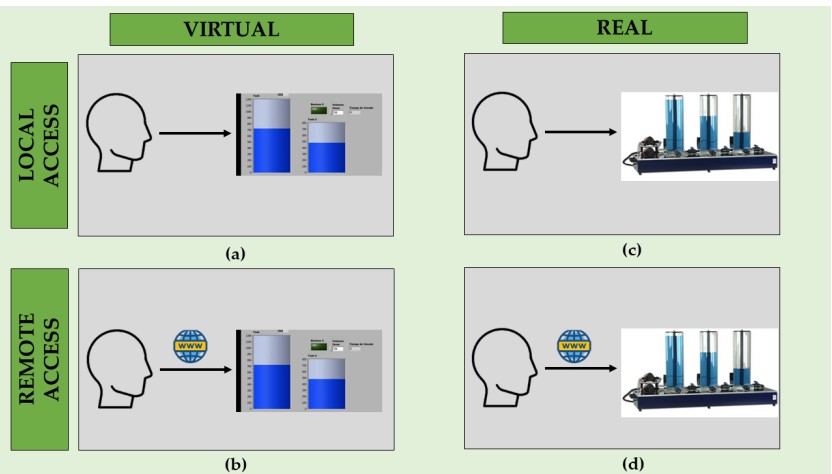

**Figure 1.** Types of laboratories: (**a**) virtual laboratory with local access; (**b**) virtual laboratory with remote access; (**c**) real laboratory with local access; (**d**) real laboratory with remote access.

The main advantages of remote laboratories are [9,10]:

- Remote laboratories can be accessed 24 h/7 days a week from an Internet connection, without requiring staff supervision.
- Due to more efficient sharing of resources, educational institutions can satisfy laboratory demand with less equipment.
- Students from universities with fewer resources can make use of high-tech laboratories.
- Students with difficulties in adjusting their work timetable with their class timetable have greater ease when conducting laboratory practices.
- The user interface is available to be run on several platforms, ranging from desktop and laptop PCs to tablets and smart phones.
- Students can also repeat the laboratory practice as many times as they desire without any extra assistance required from staff in order to gain a better understanding and further knowledge.
- Remote experiences can be shared between universities, and students have at their disposal a higher number of laboratories.
- Remote laboratories allow access to state-of-the-art experiments, avoiding risks of equipment damage and accidents.

With such advantages, many universities have decided to implement remote laboratories in their educational methodology to adapt to this coming digital era.

Summarizing the above, taking into account the advantages of remote laboratories and how the European Union (EU) aims to adapt education to the new digital era, this paper presents a solution that integrates a European collaborative network of remote laboratories based on renewable energies technology with standardized information and communications technology (ICT) infrastructure.

The next section offers a scientific literature review, revising previous related works, comparing the authors' proposal with these earlier findings, and highlighting the main contributions of the proposal. Section 3 describes the Materials and Method used to conduct the research, and results are discussed in Section 4. The main conclusions of the paper are summarized in Section 5.

## 2. Scientific Background. Related Works

Based on scientific literature, the following findings based on the development of remote laboratories in the field of engineering education deserve special attention. In 2014 the Institute of Information Engineering, Automation, and Mathematics, STU, in Bratislava [11], presented a remote laboratory to be used in process control courses. The remote experience allows interaction with three thermal plants, a magnetic levitation plant, and a hydraulic tank system. The same year, the Chiba Institute of Technology [12], implemented a remote laboratory that controls by pulse-width modulation (PWM) a direct current (DC) motor and measures the voltage of a single-phase alternator. The student needs only a web browser to access the remote laboratory, and it is available to be run in laptop, smartphone, or tablet. A more ambitious project was carried out by the University of Padova in 2015 in collaboration with industry [13]. They implemented a remote refrigeration laboratory running under the LabVIEW® environment, and the real physical plant was located at an industrial site. The real physical plant has a vapor compression cycle (VCC), and the system has four main components: evaporator, compressor, condenser, and expansion device. In an attempt to join education and inclusion, ITMO University of Saint Petersburg implemented a remote laboratory which allowed students with special education needs to gain knowledge in electrical engineering and electronics, control theory and systems identification [14]. In that work, the developed laboratory setup included DC motor monitoring and control. Students accessed the remote lab using LabVIEW® remote panels through a common web browser. More recently, in 2020, Slovak University of Technology in Bratislava described a remote laboratory in the area of control engineering [15]. This remote laboratory consists of three tank-based plants and applies the principles of nonlinear systems. The user interface is based on MATLAB®, the obtained data are transmitted to the client via WebSockets, and students access the remote lab via Modular Object-Oriented Dynamic Learning Environment (Moodle) (from Australia).

All above-described remote laboratories have been developed by single universities, without any other participating entity, except in the case of Beghi et al. (2015) [13] which included industrial collaboration.

The first example of remote laboratory network found in the scientific literature is Pastor et al. (2020) [16], where the Spanish National University for Distance Education (UNED), the German Jordanian University, and the Princess Sumaya University for Technology (PSUT) implemented a cooperative remote laboratories network. The network is built over four remote laboratories based on renewable energies: a wind and photovoltaics power trainer, where students learn the characteristics of solar photovoltaic panels and wind power generators (windmill, electric fan, solar photovoltaic panel, anemometer, temperature sensor, and load box). Students access via Moodle as a learning management system (LMS) to interact with the remote laboratory.

More recently, Northern Illinois University and the University of Illinois at Chicago developed in 2022 a virtual laboratory that controls a microgrid with renewable energy sources [17]. The laboratory simulations are performed in MATLAB/Simulink and LabVIEW®. The virtual laboratory is composed of different experimental devices: a solar tracking system, a wind turbine, and an energy storage system.

Also in 2022, a remote laboratory based on an experimental robotics course was implemented using digital twin (DT) technology, IoT technology, and the adopted analysis, design, development, implementation, and evaluation (ADDIE) teaching method. The physical plant includes an ABB IRB120 robot, and students use RobotStudio software to program and debug the robot [18]. UNIMINUTO university has implemented a low-cost and open-source remote laboratory based on tanks control to learn automatic control systems [19]. Recently, Turin Polytechnic University in Tashkent (TTPU, Tashkent, Uzbekistan) and the Politecnico di Torino (PoliTo, Italy) have developed an academic cooperation framework. The remote laboratory allow students to apply automatic control theory on a real magnetic levitation system [20]. On the other hand, the Complutense University of Madrid has implemented a remote laboratory using the Dobot Magician educational robot.

This robot is used in the practices of electronic engineering in communications and allows students to acquire knowledge about the basic principles in robotics and to detect failures in practice [21].

Based on the literature review, the authors conclude that there have been few remote laboratories developed for engineering education, and, within these few works, only one offers a network of remote laboratories, consisting of four laboratories located in different universities in Jordan. The lack of common standards in the definition and implementation of remote laboratories as well as the lack of connectivity specifications are major drawbacks to adopting and creating wide-scale networks of educational remote laboratories based on renewable energies.

In this paper, the authors propose a solution to build a complete course which integrates a standardized remote-laboratory-based network, supplementary learning material, and self-evaluation sheets, where each laboratory is located at a different place (each laboratory is located at each university). The student accesses through a common Moodle-based LMS, and after a single login step, he/she can select the module and the remote laboratory he/she wants to practice with. Table 1 compares the authors' proposal with related works and summarizes the main contributions.

**Table 1.** Comparison of design and implementation of remote laboratories in literature.

| Reference | Type of Laboratory | Application | LMS | Booking System | Standardized [3] ICT Solution | ICT Hardware and Software |
|---|---|---|---|---|---|---|
| Authors' proposal | Real laboratory with remote access [1] | Renewable energy residential | Yes | Yes | Yes | Arduino [6] + LabVIEW (UI) [7] + LMS (Moodle) [8] |
| [11] | Real laboratory with remote access | Engineering education | No | No | No | Raspberry Pi [8] + Arduino + HTML 5 [9] and JavaScript [10] (client) + PHP [11] and MySQL [12] (server) + JSON [13] (data transfer) |
| [12] | Real laboratory with remote access | Engineering education | No | No | No | HS/3069F microcomputer board + TCP and UDP + WebSocket + HTML, CSS [14] and JavaScript (UI) |
| [13] | Real laboratory with remote access | Engineering education [4] | No | No | No | VNC [15] + DAQ (NI-CompactDAQ) + LabVIEW (UI) |
| [14] | Real laboratory with remote access | Engineering education | No | No | No | NI ELVIS II + LabVIEW + TCP/IP protocol |
| [15] | Real laboratory with remote access | Engineering education | Yes | No | No | MATLAB [16] + WebSockets + JavaScript + USB interface |
| [16] | Real laboratory with remote access | Engineering education | Yes | No | Yes | LMS (Moodle) [5] |
| [17] | Virtual laboratory with remote access [2] | Engineering education | No | No | No | MATLAB/Simulink + LabVIEW + Microsoft.Net Core [17] + TCP/IP protocol + SolidWorks [18] + Inkscape [19] + Fritzing [20] + CanvasJS [21] + jQuery + JSON |

**Table 1.** *Cont.*

| Reference | Type of Laboratory | Application | LMS | Booking System | Standardized [3] ICT Solution | ICT Hardware and Software |
|---|---|---|---|---|---|---|
| [18] | Real laboratory with remote access | Engineering education | No | No | No | Raspberry Pi + RobotStudio [22] + Unity3D [23] + OPC UA protocol + MATLAB/Simulink |
| [19] | Real laboratory with remote access | Engineering education | No | No | No | Raspberry Pi 4 + Janus WebRTC Server [24] + NodeJS [25] + Flask [26] + ACE code editor and highlighter + Redis [27] |
| [20] | Real laboratory with remote access | Engineering education | No | Yes | Yes | NI MyRIO board + LabVIEW |
| [21] | Real laboratory with remote access | Engineering education | Yes | No | No | Raspberry Pi + Dobot [28] + Python [29] + EJsS [30] |

[1] A remote laboratory is defined as an environment where a student remotely controls a process and/or device through a network. Under this scheme, the student uses and controls the available resources in a laboratory by means of the use of sensors and instrumentation to undertake actual interaction with real equipment, instead of using programs that simulate the processes to be observed and studied. [2] A virtual laboratory is defined as a computer-simulated environment in which the conditions for experimentation typical of a conventional laboratory are recreated using generic or specific computer software. [3] A standardized model involves collaboration between different universities. [4] This remote laboratory is for engineering education, but the physical plant is located in an industrial site, not at a university. [5] Pastor et al. (2020) [16] do not provide further information on communication protocols or software/hardware resources used. [6] Arduino (Italy). [7] LabVIEW (United States). [8] Raspberry Pi (United Kingdom). [9] HTML5 (Switzerland). [10] JavaScript (United States). [11] PHP (PHP Group). [12] MySQL (United States). [13] JSON (United States). [14] CSS (Switzerland). [15] VNC (United Kingdom). [16] MATLAB (United States). [17] Microsoft.Net Core (United States). [18] SolidWorks (France). [19] Inkscape (United States). [20] Fritzing (Germany). [21] CanvasJS (Australia). [22] RobotStudio (Switzerland). [23] Unity3D (Denmark). [24] Janus WebRTC Server (Italy). [25] NodeJS (United States). [26] Flask (United States). [27] Redis (Italy). [28] Dobot (Spain). [29] Python (United States). [30] EJsS (Spain).

The main novelties of the proposal are:

1. A collaborative training platform based on standardized remote laboratories is developed. The training platform is made up of five EU universities.
2. The application: the remote laboratory is based on renewable energy with the aim of contributing to training skills required from students in the new energy model.
3. An LMS using a cooperative Moodle platform. All participating universities are able to create content and teaching material for the cooperative training platform.
4. A booking system: with this tool, the student selects the date and time to carry out the remote experience.
5. A standardized ICT solution that can unify the criteria for the five participant universities. All higher institutions have followed the same ICT standards. This provides replicability and scalability and makes it easy to add new devices at the laboratory and incorporate new universities in the collaborative network.
6. ICT hardware and software: Arduino + LabVIEW (UI) + Enlarge/MyFrontier + Moodle platform.

## 3. Materials and Method

### 3.1. REOPEN Approach. Context and Task Distribution

The remote laboratories for practical experiments on renewable energies at EU universities (REOPEN) approach provides easier access for students to scientific–technological laboratory experiments when the student is away from the university campus. The result is a collaborative platform made up of standardized remote-laboratory-based training modules developed by EU universities. The aim is to create a new educational model with the implementation of remote laboratories, merging traditional and online learning. The

remote laboratories are based on renewable energies with the aim of contributing to those training skills in energy transition required from students.

The European universities participating in the REOPEN project are Guglielmo Marconi University (Italy), University of Applied Sciences Technikum Wien (Austria), Munster Technological University (Ireland), Norwegian University of Science and Technology (Norway) and the University of Huelva (Spain). Each of these universities is in charge of one intellectual output (IO), as shown in Figure 2.

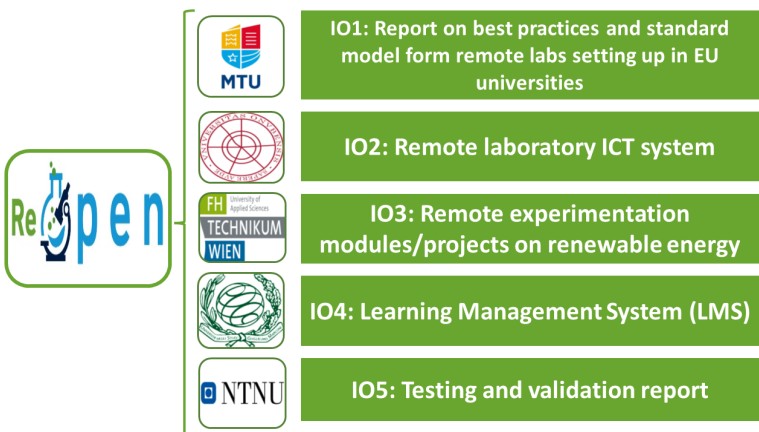

**Figure 2.** Intellectual outputs (IOs) of each European university.

Munster Technological University (MTU) oversees IO1 to collect experiences in development and use of hands-on, virtual, and remote laboratories. Additionally, MTU is in charge of analyzing the use of remote laboratories in Engineering and science, technology, engineering, and mathematics (STEM) careers and designing a pedagogical model of remote laboratories. The consortium has distributed surveys to professors (Figure 3a), students (Figure 3b), and industry workers (Figure 3c) between the five member countries. The results obtained from these surveys are detailed in [22].

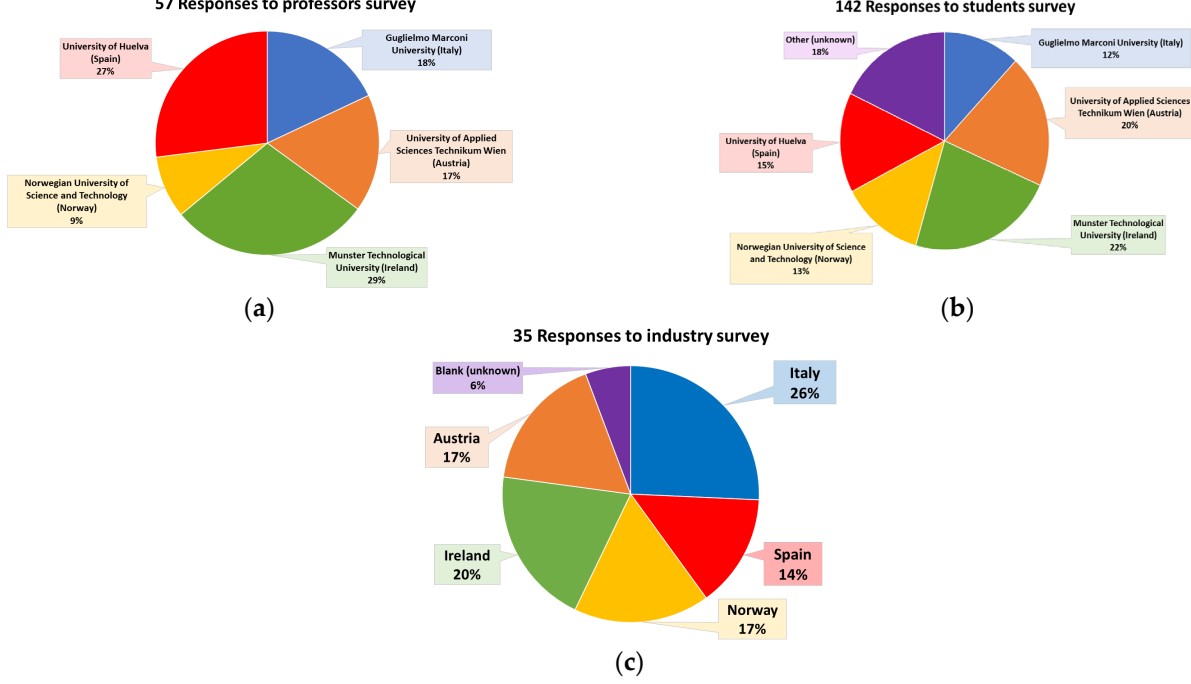

**Figure 3.** Survey data collection: (**a**) Responses to professors' survey; (**b**) Responses to students survey; (**c**) Responses to industry survey.

One of the objectives of these surveys was to find out how motivating students, professors, and industry workers found the laboratory experience, their levels of enjoyment with the laboratory, and their overall satisfaction and levels of interest in the laboratory. Figure 4 shows how the University of Huelva (UHU) achieved the highest degree of satisfaction (4.5/5).

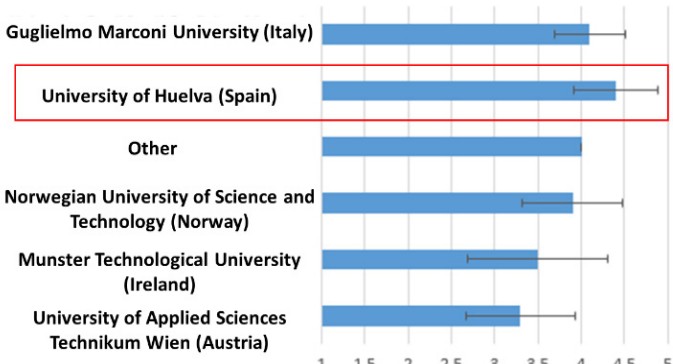

**Figure 4.** Satisfaction level with laboratory practices. Results from surveys of professors, students, and technical staff.

On the other hand, the University of Huelva (UHU) is responsible for IO2, defining the standard ICT for the remote laboratory, establishing the communication system, including the use of programmable devices for the remote connection of the experimental devices, establishing the harmonized and standardized remote lab infrastructure, and the elaboration of guidelines for the user. The implementation of the remote laboratory ICT infrastructure is based on the architecture shown in Figure 5. The user, who must have a computer with Internet connection (item 1, Figure 5), accesses the REOPEN Moodle platform (item 2, Figure 5), where the five remote laboratories are hosted. Next, the REOPEN Moodle platform accesses the university's local area network (LAN) through a server (item 3, Figure 5). The intranet and server are located at the university, where the experiment hardware is located. A computer is also connected to the university's LAN. The computer (item 4, Figure 5) can be a personal computer (PC), a mini-PC, a Raspberry Pi, or a similar device that allows the installation of an operating system (OS) and has an Internet connection. Thus, the user interface (UI) is a software tool, for example a LabVIEW® file, that runs on the computer. The user accesses the UI via the Internet and, therefore, can operate the physical plant (item 5, Figure 5). The last element in the architecture is the physical plant. In the scheme, the plant refers not only to the renewable-based device but also all the necessary elements for the experiment automation. A proper control and monitoring system is key, gathering the functions of operating the plant, monitoring its status, and exchanging data with the UI; that is, it is connected to the minicomputer. A detailed description of the remote laboratory implementation is given in Section 3.2.

As authors have commented, the proposed educational model and the intended EU collaborative network are built over a common basis: a renewable-based residential microgrid. Institutions have selected different remote laboratories related to different renewable energy sources to be part of the residential microgrid. As an example, this paper describes the remote laboratory developed by the University of Huelva. The ICT and infrastructure are replicated by the rest of the entities.

After the surveys and the remote laboratory implementation, University of Applied Sciences Technikum Wien (FHTW) is in charge of designing the experimentation modules on renewable energies developed by each participant university and writing the curriculum of these modules (IO3). Guglielmo Marconi University (USGM) is responsible for designing the LMS (IO4), which includes the training course and remote laboratories of each European university. Norwegian University of Science and Technology (NTNU) has to select the students who carry out the different practical experiences and elaborate the guidelines that the users have to follow in order to carry out the experimental activities (IO5).

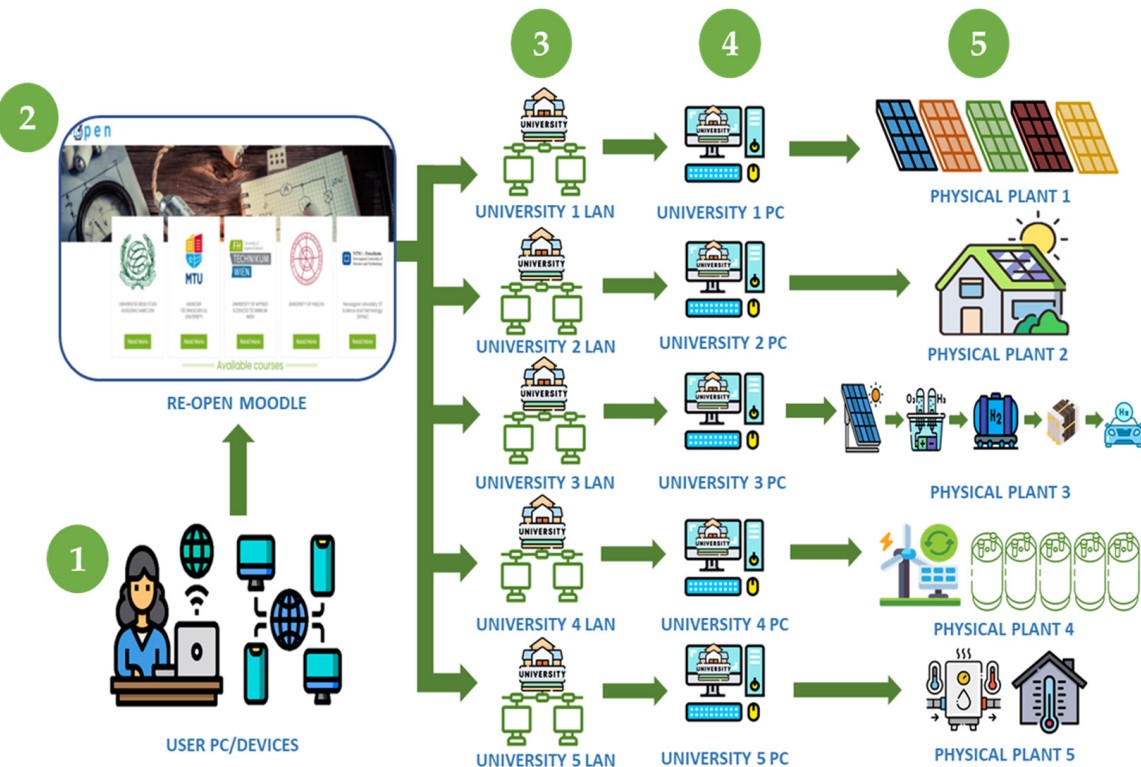

**Figure 5.** REOPEN ICT infrastructure standardization and collaborative network design.

### *3.2. Standardized Remote Laboratory. An Example Based on Supercapacitor Power Bank*

In the context of the REOPEN project, the UHU has implemented a remote laboratory based on a supercapacitor power bank (physical plant 4, Figure 5). Supercapacitors are high-capacity and have a capacitance value much higher than other capacitors but with lower voltage limits, bridging the gap between electrolytic capacitors and rechargeable batteries. With this advantage, supercapacitors enable fast charging and discharging. The aim of this remote laboratory is to allow students to interact with a real plant through a user interface, visualizing the charging and discharging curves of the supercapacitor power bank. The user interface allows the student to visualize the supercapacitor power bank in real time. That is, a laboratory practice similar to a face-to-face laboratory is provided but with the novelty that student is training from home.

### 3.2.1. System Description

The supercapacitor power bank (item 1, Figures 6 and 7) is connected through a single-pole double-throw (SPDT) relay (item 2, Figures 6 and 7) to choose between the charging or discharging circuits.

For the charging circuit, the supercapacitor power bank can be charged from a programmable power source (item 3, Figures 6 and 7), according to technical specifications, Table 2. A protection fuse is placed for the charging process (item 4, Figures 6 and 7) to avoid potential risks.

For the discharging circuit, an electronic load (item 5, Figures 6 and 7) is used. The electronic load consists of a bank of 8 power resistors connected in parallel. Each resistor is driven by a relay.

**Table 2.** Supercapacitor power bank. Component specifications.

| Component | Parameter | Units |
|---|---|---|
| Supercapacitor | Capacity: 5000 F<br>Surge voltage: 2.85 VDC<br>Operating voltage: 2.4 VDC<br>Max. Continuous current: 210 A | 5 |
| Relay SPDT | Max. Switching current: 80 A<br>Max. Switching Voltage: 75 VDC<br>Rated voltage (VDC): 12 V | 1 |
| Programmable power source | Input voltage: 220 VAC<br>Output voltage: 0–30 VDC<br>Output current: 0.1–100 A<br>Output power: 3000 W | 1 |
| Fuse | Rated current: 100 A<br>Rated voltage: 500 VAC | 1 |
| Resistor power bank | Resistance: 1 $\Omega$<br>Power rating on standard heatsink @25 °C: 200 W | 8 |
| Relay module | Normally open interface maximum load: 250 VAC/30 A; 30 VDC/30 A<br>Trigger current: 3 mA<br>Working voltage: 12 VDC | 8 |
| Current sensor | Primary nominal RMS current: 100 A<br>Primary current, measuring range: 0, . . . , $\pm$150 A<br>Secondary nominal RMS current: 50 mA | 1 |
| Switching power supply | Input voltage range: 88~264 VAC<br>Output voltage: 12 VDC<br>Output current range: 0~4.2 A<br>Output rated power: 50.4 W | 1 |
| Voltage sensor (implemented by resistor divisor) | Range: 0–12 VDC | 1 |

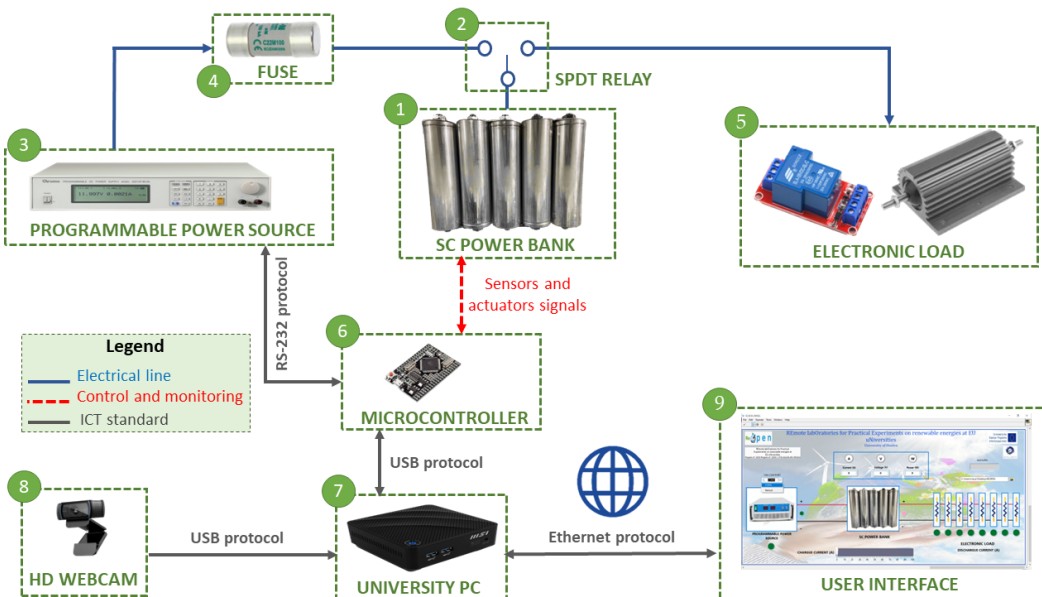

**Figure 6.** Supercapacitor remote laboratory. Component layout.

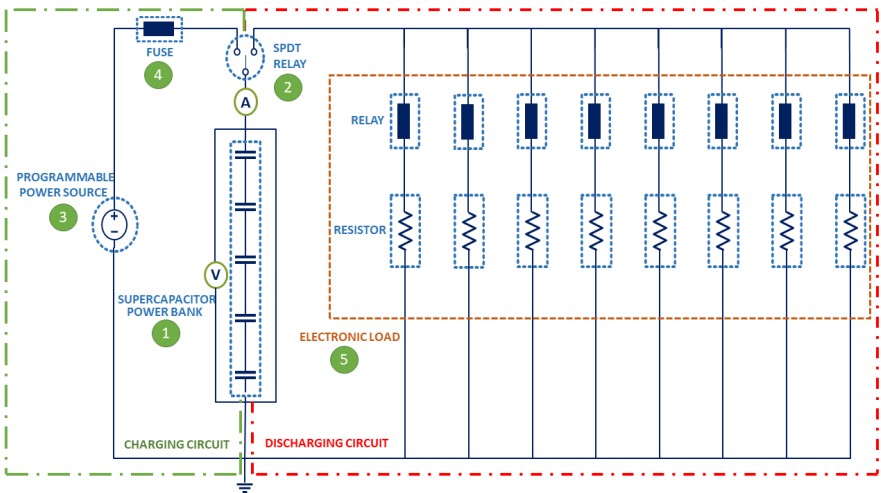

**Figure 7.** Supercapacitor test bench. Electrical layout.

For the control and data acquisition system, an Arduino microcontroller (item 6, Figure 6) is used. The Arduino microcontroller is connected via a Universal Serial Bus (USB) interface to a local computer (item 7, Figure 6) and via the RS-232 protocol to a programmable power source. The student can visualize the laboratory physical plant in real time through a USB-connected video camera (item 8, Figure 6). Similarly, the information obtained by Arduino is displayed through a UI (item 9, Figure 6) at the user PC connected remotely. This user interface allows the student to interact with the real plant.

The technical specifications of each component used in the supercapacitor test bench are shown in Table 2.

The physical implementation of the supercapacitor power-bank-based remote laboratory is shown in Figure 8. The image on the left shows the charging process of the supercapacitor power bank, and the image on the right shows the process of the discharging cycle.

For the charging and discharging tests, the operating conditions to be imposed for security and avoiding risk in remote mode are shown in Table 3.

**Table 3.** Supercapacitor power bank system specifications.

| Parameter |
| --- |
| Operating voltage 1-single supercapacitor module: 2.4 VDC |
| Operating voltage supercapacitor power bank: 12 VDC |
| Charging current: 0~70 A |
| Discharging current: 0~96 A |

### 3.2.2. ICT Infrastructure and Learning Management System

The developed educational structure encompasses the remote laboratories of the participant universities, fulfilling the objective of creating an infrastructure of remote and collaborative laboratories in EU universities, as shown in Figure 9. All remote laboratories are accessible from Moodle. The integration was carried out with Shareable Content Object Reference Model (SCORM). The decision to select SCORM as the content sharing tool was based on the recommendation of the company VJTechnology®, which has provided technical support to REOPEN partnership. As extra features, the LMS facilitates the reservation of remote laboratories through a booking system and interactive material such as augmented reality (AR) interactive videos.

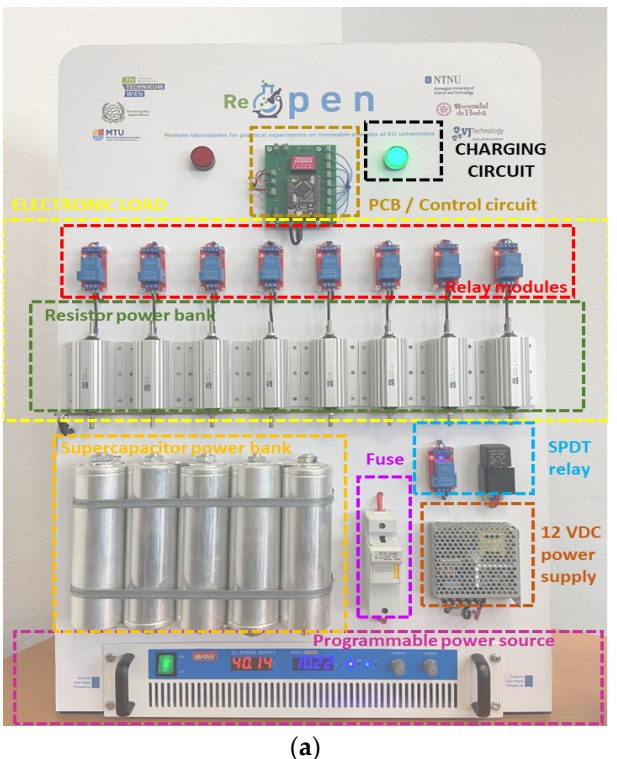
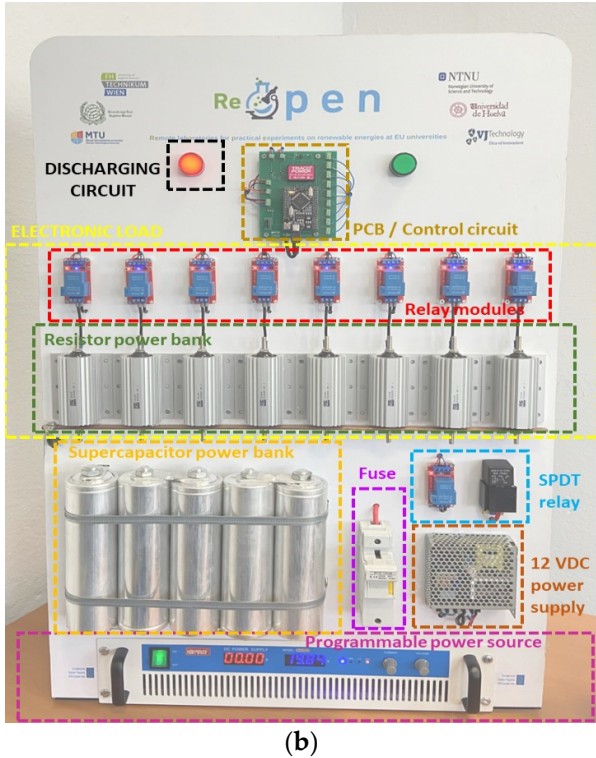

**Figure 8.** Physical implementation of the remote laboratory based on supercapacitor power bank: (**a**) charging process; (**b**) discharging process.

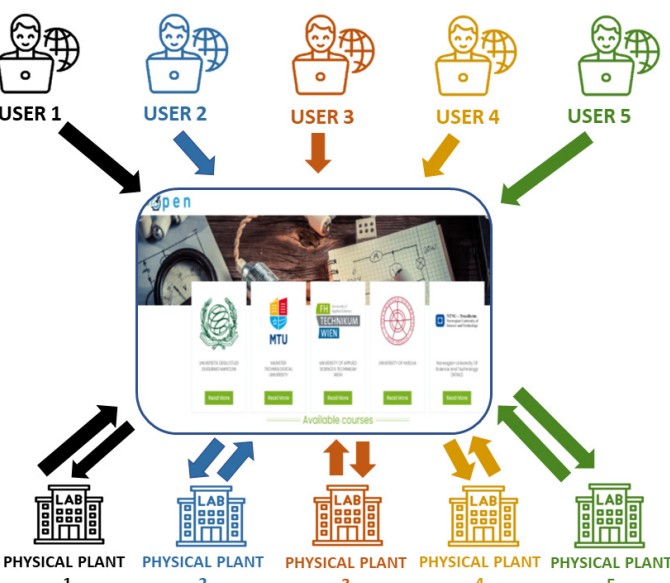

**Figure 9.** LMS of REOPEN platform.

While one user is remotely accessing a laboratory, the rest of the plants are available for other users. It is thus possible to conduct five remote experiences at the same time.

To validate the student's authentication, the student logs into the university portal. An identification parameter is then be sent to REOPEN Moodle, which verifies their presence in its database. The access to LMS is established by means of identity and access management (IAM) [23]. IAM confirms that the user, software, or hardware is authentic by using the REOPEN LMS to authenticate the credentials in a database, as shown in Figure 10.

**AUTHENTICATION**

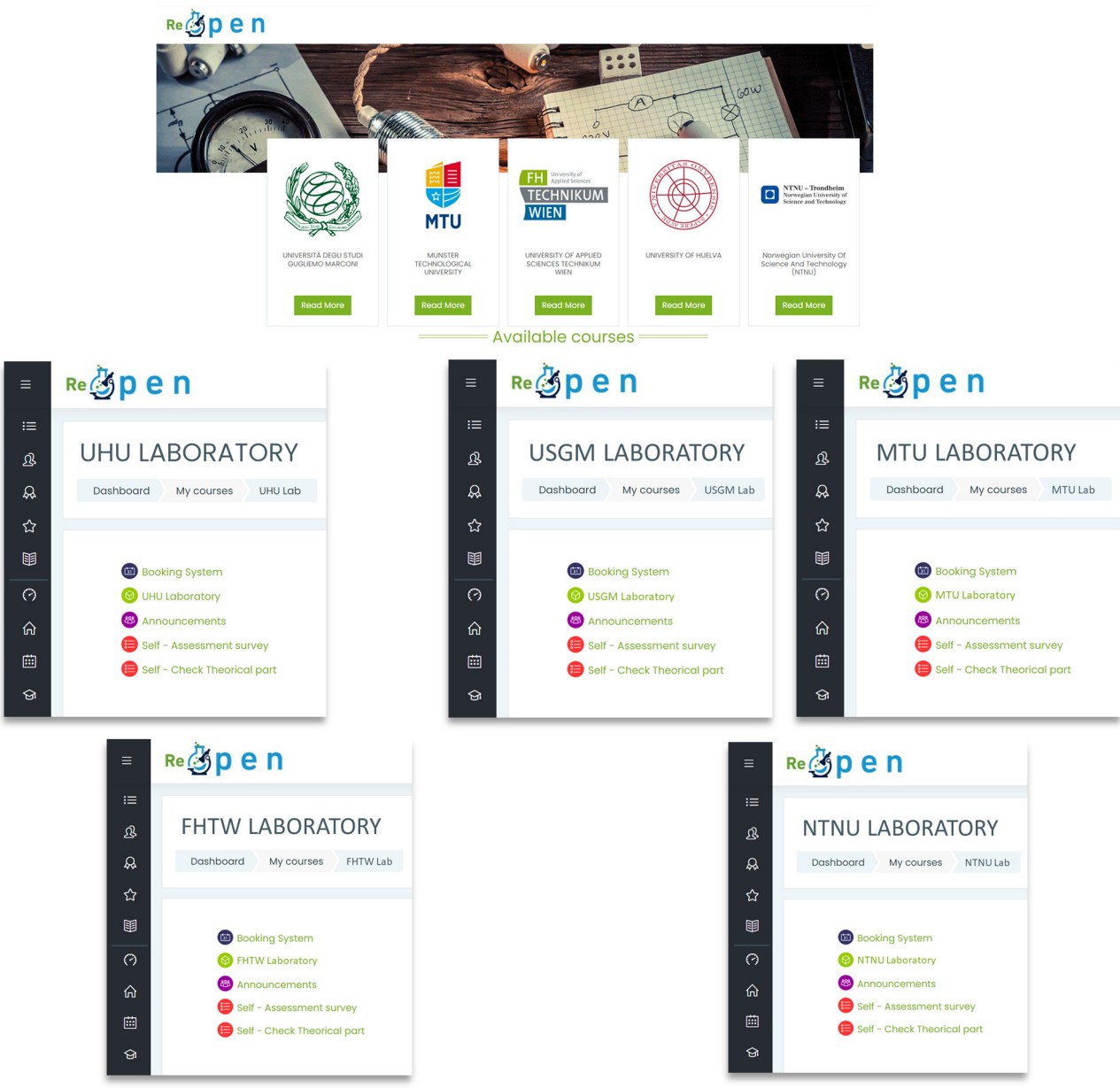

**Figure 10.** Authentication process in REOPEN platform.

In order to realize this standardized and collaborative structure between the five participating European universities, a common LMS has been implemented for all universities. This LMS, namely a Moodle platform, incorporates the five remote experiences of each university. To access the REOPEN platform, students can access the following URL, https://reopen.vjtechnology.it/index.php (accessed on 14 April 2023), as shown in Figure 11. The student must be enrolled in a course at one of the five European universities.

**Figure 11.** REOPEN LMS platform.

Then, the student must enter the username and password. The username and password are provided by the professor of the course at the beginning of the academic year, thus ensuring the correct functioning of the collaborative platform in terms of security.

Once the student logs into REOPEN LMS, the different modules with respective remote laboratories are shown. The user has access to the booking system, including the remote experience, announcements, self-assessment sheets, and self-check theoretical section. Before starting the remote experience, the professor gives a lecture on renewable energies. The first thing the student should do before the remote experience is to fill in the self-check theoretical section. This survey asks about the theoretical knowledge explained in class. Once the student has completed the remote experience, the student can carry out an assessment of the knowledge acquired filling the "self-assessment survey". This survey is subsequently evaluated by the professor.

### 3.2.3. User Interface

Once inside the course, for example the "UHU Laboratory" tool, the student can access the remote laboratory implemented by the UHU. As shown in Figure 12, the remote laboratory consists of the user interface at the bottom, and at the top a video shows in real time the operation of the physical plant.

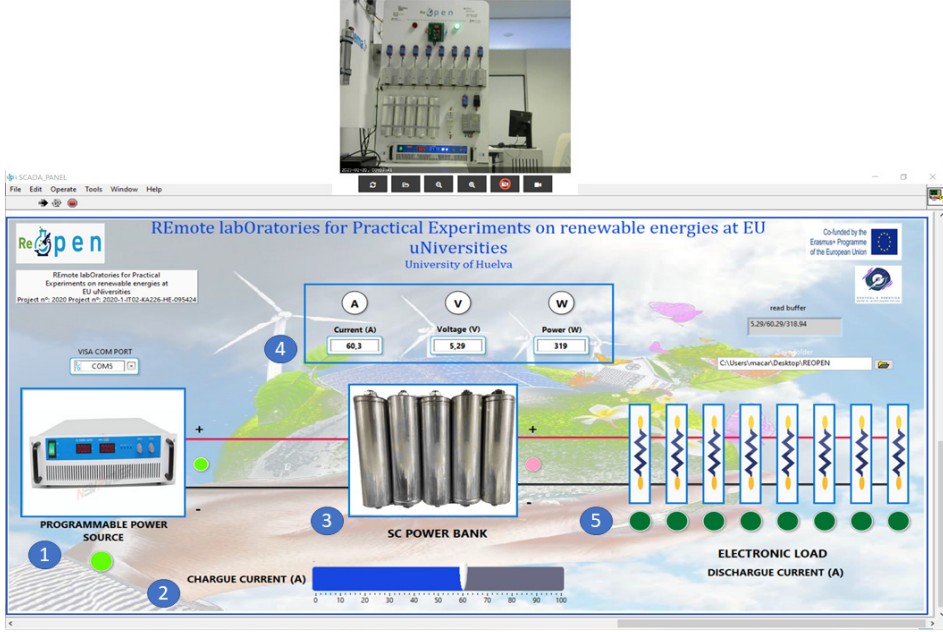

**Figure 12.** User interface of the remote laboratory of supercapacitor power bank: charging process.

The user interface is created in the LabVIEW® environment. To start with the remote practice, firstly the student must turn on the programmable power source by clicking on the image (item 1, Figure 12). Once the programmable power source is turned on, the student must select the charging current of the supercapacitor power bank. To select the charge current, the student must move the slider bar (item 2, Figure 12). Then, the charging process of the supercapacitor power bank starts. If the student clicks on the image of the supercapacitor power bank (item 3, Figure 12), a second window can be seen by the student, as shown in Figure 13, and the student can visualize the different charging curves. In the upper part of the main window in the user interface, the student can visualize the instantaneous current, voltage, and power values of the supercapacitor power bank (item 4, Figure 12). Once the voltage at the supercapacitor power bank reaches 12 voltage direct current (VDC), the charging process stops. Based on the remote laboratory implementation, it is possible to carry out open-ended laboratory practices.

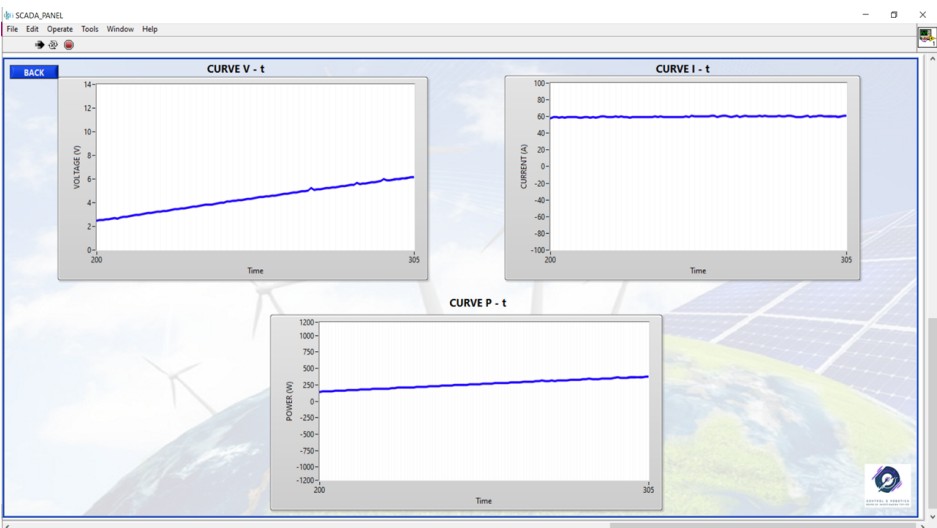

**Figure 13.** Second window in user interface. Charging and discharging chart of the supercapacitor power bank.

Similarly, the student can carry out an experimental test to draw the discharging process. To do this, the student must turn off the programmable power source (item 1, Figure 14) and tun on the desired number of resistors (item 5, Figure 14). The discharging current is conditioned by the number of selected resistors. The more resistors selected, the higher the discharging current. In the same way as for the charging process, the student can monitor the instantaneous current, voltage, and power values at the top on the main window of the user interface (item 4, Figure 14). The student can also monitor the discharging curves in the second window of the user interface, as shown in Figure 13.

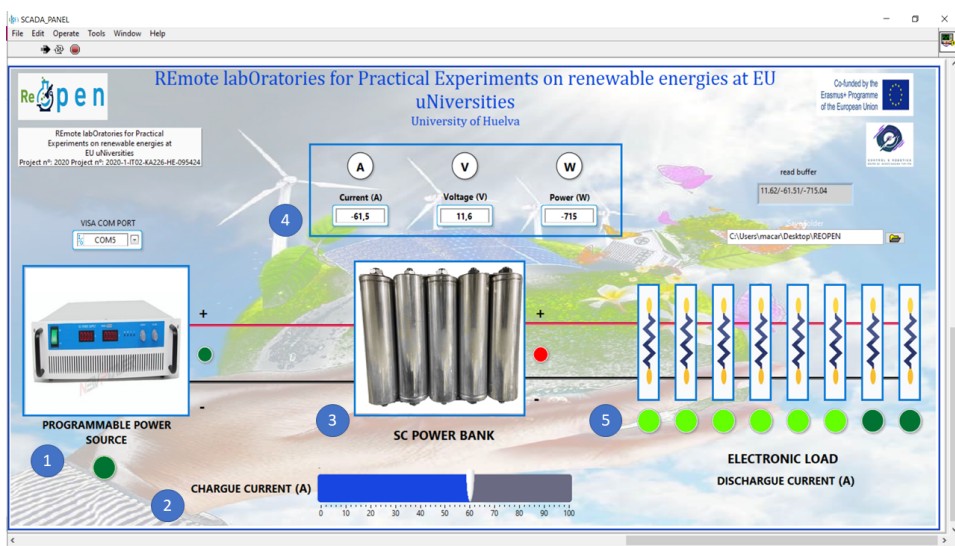

**Figure 14.** User interface of the remote laboratory of supercapacitor power bank: discharging process.

## 4. Results and Discussion

As already mentioned, the UHU, together with four other European universities, have developed a standardized remote laboratories infrastructure. To show the contributions and capabilities of the developed collaborative platform, a statistical study was conducted, as shown in Figure 3. The statistical study is based on an anonymous expectation survey that was filled in before the remote experience. This initial survey includes questions to find out whether the students and professors have had previous experience with remote laboratories. Once the students have completed the remote experience, they fill in a survey on technical

knowledge acquired. Afterwards, users have completed an anonymous satisfaction survey. The questions were asked to rate on a 5-point Likert scale (1: strongly disagree and 5: strongly agree). The results of the analysis in terms of the objectives of this study are presented below.

### 4.1. Expectation Survey

The aim of this survey is to find out whether students and professors have had previous remote experiences and their opinion on the use of these new technologies. As can be seen in Tables 4 and 5, few students and professors have previously had the opportunity to use remote laboratories in the teaching–learning method. Similarly, students and professors have heard about this or seen videos about remote laboratories, but they have not actively participated. Both students and professors think that the use of remote laboratories enhances the learning method. Finally, a large majority of them think that remote labs should be included within the curricula plans.

**Table 4.** Students' responses to expectation survey.

| Q1 **Have you previously had the opportunity to gain experience with remote laboratories?** | Yes (%) | | | No (%) | |
|---|---|---|---|---|---|
| | 20 | | | 80 | |
| Q2 **If Yes on the previous question, what kind of experiences did you acquire about the remote laboratories?** | I have heard about this but have not actively participated (%) | I have seen a video on internet but have not actively participated (%) | | I had one remote laboratory (%) | I have had several such remote laboratories in my education (%) |
| | 40 | 35 | | 20 | 5 |
| Q3 **Do you think it would be or it was useful for your study to have experience with remote laboratories?** | It is not at all useful (%) | Little useful (%) | Partially useful (%) | Useful (%) | Very useful (%) |
| | 2 | 3 | 5 | 60 | 30 |
| Q4 **Do you think that the remote laboratories should be included more in the curricula plans?** | Yes (%) | No (%) | | Unsure (%) | |
| | 72 | 8 | | 18 | |

As we can see from the data collected in Table 4, 80% of participants did not have any previous experience with remote laboratories. Nevertheless, 90% of students consider remote laboratories useful or very useful, and 72% of participants recommend their inclusion in the curricula plans.

From Table 5, 73% of professors did not have any previous experience with remote laboratories. Similar to students, despite not having previous experience, a large percentage, 85%, of professors consider remote laboratories useful or very useful. Nevertheless, professors are not yet convinced that remote labs should be included in the curricula.

**Table 5.** Professors' responses to expectation survey.

| Q1 Have you previously had the opportunity to gain experience with remote laboratories? | Yes (%) | | | No (%) | |
|---|---|---|---|---|---|
| | 27 | | | 73 | |

| Q2 If Yes on the previous question, what kind of experiences did you acquire about the remote laboratories? | I have heard about this but have not actively participated (%) | I have seen a video on Internet but have not actively participated (%) | | I had one remote laboratory (%) | I have had several such remote laboratories in my education (%) |
|---|---|---|---|---|---|
| | 30 | 35 | | 25 | 10 |

| Q3 Do you think it would be or it was useful for your teaching to have experience with remote laboratories? | It is not at all useful (%) | Little useful (%) | Partially useful (%) | Useful (%) | Very useful (%) |
|---|---|---|---|---|---|
| | 4 | 5 | 6 | 50 | 35 |

| Q4 Do you think that the remote laboratories should be included more in the curricula plans? | Yes (%) | No (%) | | Unsure (%) | |
|---|---|---|---|---|---|
| | 66 | 12 | | 12 | |

### 4.2. Self-Assessment Survey

The professor assessed the technical knowledge acquired by the students during the remote experience. Prior to this, the students filled in the "self-assessment survey". In this survey, they were asked several questions about the technical knowledge they had applied in order to be able to carry out the remote experience. The questions asked in the "self-assessment survey" are shown in Table 6. The professor evaluated the questions with a percentage of correct or incorrect questions.

**Table 6.** Responses from students to self-assessment survey.

| | | | Correct (%) | Incorrect (%) |
|---|---|---|---|---|
| Q1 | What is the maximum charging current? | | 40 | 60 |
| Q2 | What is the maximum charging voltage of the supercapacitor power bank? | | 73.33 | 26.67 |
| Q3 | How many supercapacitors are there? | | 80 | 20 |
| Q4 | How many resistors are there? | | 86.67 | 13.33 |
| Q5 | What is the maximum discharging current? | | 53.33 | 46.67 |
| Q6 | How many variables can you plot in the graphs? | | 66.67 | 33.33 |
| Q7 | When the supercapacitors power bank is charged, do you have to turn off the programmable power source? | | 93.33 | 6.67 |
| Q8 | After the experiment, what types of values are saved in the txt file? | | 60 | 40 |

### 4.3. Satisfaction Survey

In general, it can be said that the students and professors have had a very high acceptance of the integration of the remote laboratories. This acceptance is due to the fact that remote laboratories offer versatility and flexibility for learning. The versatility and flexibility are due to the fact that remote laboratories do not depend on a predetermined schedule and allow the experiment to be repeated as many times as students require. In

fact, some students and professors expressed their excitement at being able to try out the other remote laboratories integrated into the standardized REOPEN platform.

Based on students' and professors' answers, shown in Tables 7 and 8, it can be seen that students and professors consider being able to access remote labs from other universities useful (Table 7, Q0; AVG_Students = 4.8; AVG_Proffessors = 4.82, and Table 8, Q5; AVG_Students = 4.37; AVG_Professors = 4.14).

**Table 7.** Usefulness of remote laboratories according to the students and professors.

| | | Students | | Professors | |
|---|---|---|---|---|---|
| | | **AVG** | **SD** | **AVG** | **SD** |
| **Q0** | Being able to access other remote labs from other universities is useful | 4.80 | 0.79 | 4.82 | 0.81 |
| **Q1** | The use of remote laboratories helps reinforce theoretical concepts | 4.25 | 0.64 | 4.29 | 0.73 |
| **Q2** | The remote laboratory helps to understand the importance of the use of renewable energies | 3.88 | 0.58 | 4.01 | 0.66 |
| **Q3** | The use of remote laboratories is safer than the use of physical laboratories | 3.92 | 0.62 | 3.74 | 0.54 |
| **Q4** | The overall assessment of the remote lab is positive | 4.40 | 0.58 | 4.53 | 0.72 |

AVG: Average. SD: Standard deviation.

**Table 8.** Usability perceived by students and professors.

| | | Students | | Professors | |
|---|---|---|---|---|---|
| | | **AVG** | **SD** | **AVG** | **SD** |
| **Q5** | The remote laboratory allows the activity to be carried out in the same way as the physical laboratory | 4.37 | 0.49 | 4.14 | 0.55 |
| **Q6** | The activity can be carried out without supervision | 4.29 | 0.60 | 4.02 | 0.62 |
| **Q7** | Time flexibility helps to carry out the activity | 4.75 | 0.54 | 4.73 | 0.50 |
| **Q8** | The user manual provided in the LMS helps to carry out the activity | 3.81 | 0.80 | 4.26 | 0.78 |
| **Q9** | Using the booking system is easy | 4.11 | 0.69 | 4.79 | 0.77 |
| **Q10** | Access to the remote lab via the LMS is easy | 4.29 | 0.72 | 4.54 | 0.73 |
| **Q11** | The user interface is easy and intuitive | 4.23 | 0.57 | 4.62 | 0.62 |
| **Q12** | The integration of the video camera in the remote laboratory helps to understand the activity | 3.77 | 0.51 | 4.80 | 0.74 |

AVG: Average. SD: Standard deviation.

Regarding the learning procedure (Table 7, Q1 and Q2), the students and professors confirm, by means of high scores, that remote laboratories strengthen and improve theoretical concepts (Q1; AVG_Students = 4.25; AVG_Professors = 4.29). In addition, the standardized infrastructure of REOPEN improves the understanding of the importance of the use of renewable energies (Q2; AVG_Students = 3.88; AVG_Professors = 4.01). Likewise, students and professors consider that the use of remote laboratories increases the safety of students and equipment with respect to physical laboratories (Table 7, Q3; AVG_Students = 3.92; AVG_Professors = 3.74). In general, the students and professors consider that the remote lab is a positive innovation (Table 7, Q4; AVG_Students = 4.40; AVG_Professors = 4.53).

Regarding the performance of the experiment (Table 8, Q5, Q6, Q7, and Q8), students have a better assessment (Q5; AVG_Students = 4.37) than professors (Q5; AVG_Professors = 4.14).

In addition, students believe that the activity can be performed without the supervision of a professor (Q6; AVG_Students = 4.29), while professors believe that there should be a supervisor (Q6; AVG_Professors = 4.02). In this sense, a great advantage of the remote laboratories over face-to-face laboratories is shown: the control by software substitutes the permanent teacher supervision that in face-to-face practice would be required to guarantee the correct use of lab devices.

On the other hand, students (Q7; AVG_Students = 4.75) and professors (Q7; AVG_Professors = 4.73) consider the flexibility of time available to take the tests to be a great advantage. While professors believe that the teaching material helps to learn the activity (Q8; AVG_Professors = 4.26), students were not in agreement with respect to the suitability of instructions to perform the lab work (Q8; AVG_Students = 3.81; SD = 0.80).

Professors rate the "booking system tool" (Q9; AVG_Professors = 4.79), the access to the LMS (Q10; AVG_Professors = 4.54), and the user interface (Q11; AVG_Professors = 4.62) very positively, saying that it is easy to use and intuitive. Most innovative for teachers is the implementation of a video camera that transmits the remote practice in real time (Q12; AVG_Professors = 4.80).

These same aspects were evaluated by students, who say that the "booking system tool" of the remote laboratory is easy to use (Q9; AVG_Students = 4.11), as well as remote access through the LMS (Q10; AVG_Students = 4.29). Students consider the user interface to be intuitive (Q11; AVG_Students = 4.23), although they do not perceive any real advantages of using the camera (Q12; AVG_Students = 3.77).

Remarkably, it can be observed that scores given by students to questions Q2 (Table 7), Q8, and Q12 (Table 8) were significantly lower than the scores given by professors. This may be because students still prefer face-to-face practice in subjects with a high level of experimentation such as renewable energy.

## 5. Conclusions

This work aimed to contribute to the digital transition in higher education by creating a standardized educational model based on remote laboratories focused on renewable energy and developed in an EU framework. The aim was to establish, test, and validate a standardized educational model and an ICT solution for conducting laboratory activities remotely, merging online and traditional learning and allowing students to carry out practical activities/experiments integrated with theoretical lessons on renewable energy topics.

The collaborative European network consists of five European universities that have proposed to bridge the lack of common infrastructures between different remote laboratories. The result has been the creation of the REOPEN educational network. REOPEN offers real learning opportunities to a more diverse and broader engineering student body by providing more flexible access to physical experiments anytime, anywhere. With this collaborative infrastructure, REOPEN promotes the reuse of programming codes, avoiding the fragmentation caused using proprietary tools by each university. Thus, the REOPEN network provides students with the necessary competences to meet the needs of Industry 4.0.

Initially, this work had the initial objective of interviewing at least 15 professors and 25 students from STEM faculties; 5 university staff such as laboratory technicians; and 5 business administrators for the mapping phase. In total, 57 professors' surveys and 142 students' surveys from STEM faculties, 13 surveys from university staff such as laboratory technicians, and 35 business administrators' surveys were obtained.

The data obtained from the surveys show that the implementation of remote laboratories in the teaching–learning method is a real novelty. In general, students and professors think that the incorporation of this new method is very beneficial, especially for laboratory practices. These laboratories offer a great deal of flexibility and versatility, and both students and professors have a very positive assessment of these advantages, as they allow the experiment to be repeated as many times as the students require, reinforcing the theoretical concepts. Another very positive evaluation obtained from the surveys is that the collaborative network is based on renewable energies, adapting to the new European

energy model. Above all, however, the idea of creating a collaborative network of remote laboratories with the participation of five European universities has received the most positive evaluation. This idea fills a gap that existed until now, allowing the development of new skills techniques and enabling students to broaden their theoretical and practical knowledge by being able to use more laboratories than only those that are available at their own university.

Since COVID-19 arrived in 2020, it has been transforming people's lives. REOPEN is a project that was created to cover a need that was arising at that time to continue teaching, especially at a practical level, remotely for the safety of students and professors. Since then, the teaching–learning method has been adapting to this new situation, with online teaching, i.e., hybrid education, taking center stage. Therefore, more and more remote laboratories are being incorporated in universities, especially in universities related to the scientific–technological field.

It can be said that this collaborative network of European remote laboratories is a pioneering approach to covering this new context, and in an emerging and booming field, such as renewable energies, which is a rather expensive field, not all universities can afford to have renewable energy-based laboratories inside their facilities.

To conclude, it is convenient to discuss the limits of remote laboratories. These should not be perceived as a substitute for conventional labs (on-site) but as a complement and solution to their restrictions. The remote lab, like the virtual one, is a technological tool composed of software but, in addition and unlike the virtual one, it is also composed of hardware. The remote lab allows teachers and students to carry out, through the Internet, their practice as if they were in a conventional lab (on-site). Users use and control the resources available through the use of sensors and instrumentation that allow them to interact with real equipment instead of using simulations and without requiring physical presence in the lab. Users can therefore become familiar with interfaces that correspond to real equipment that they operate remotely. An outstanding advantage of remote labs is reducing space and time limitations, as they do not require the physical presence of the user in the lab. In this way, the labs are available beyond school hours or scheduled classroom practice, which allows students to increase the number of hours of practice, an essential requirement for learning in engineering careers. This encourages autonomy in learning, since it does not require concurrence with the teacher. Another aspect to highlight is the possibility of repeating the same practice or the implementation of variations in the initial practice according to the needs and interests of the students, contributing to personalized learning and respecting the individual needs of each user.

That said, remote labs should not be seen as a substitute for conventional labs but as a complement. Consequently, it is necessary to make rational use of remote laboratories in conjunction with conventional laboratories to provide students with the most complete set of skills in the new digital era.

**Author Contributions:** Conceptualization, F.S. and J.M.A.; Methodology, F.S.; software, M.M.; validation, M.M. and F.S.; formal analysis, M.M.; resources, F.S. and J.M.A.; writing—original draft preparation, M.M.; writing—review and editing, F.S.; project administration, F.S.; funding acquisition, J.M.A. All authors have read and agreed to the published version of the manuscript.

**Funding:** This research was funded by Erasmus+ Programme, grant number Ref. 2020-1-IT02-KA226-HE-095424 RE-OPEN project; ERASMUS+ Programme 2020—KA2; and the APC was funded by Ref. 2020-1-IT02-KA226-HE-095424 RE-OPEN project, founded by ERASMUS+ Programme 2020—KA2.

**Institutional Review Board Statement:** Ethical review and approval were waived for this study due to practices with remote laboratories have been carried out inside the facilities of universities participant in the project, within the classes lessons and with internal personnel.

**Informed Consent Statement:** Informed consent was obtained from all subjects involved in the study.

**Data Availability Statement:** The data presented in this study are available on request from the corresponding author. The data are not publicly available due to internal agreement between project partnership.

**Conflicts of Interest:** The authors declare no conflict of interest.

**List of Acronyms**

| | |
|---|---|
| AR | Augmented reality |
| DC | Direct current |
| EU | European Union |
| FHTW | University of Applied Sciences Technikum Wien |
| IAM | Identity and access management |
| ICT | Information and Communications Technology |
| LAN | Local area network |
| LMS | Learning management system |
| MOODLE | Modular Object-Oriented Dynamic Learning Environment |
| MS | Member states |
| MTU | Munster Technological University |
| NTNU | Norwegian University of Science and Technology |
| OS | Operating system |
| PC | Personal computer |
| PSUT | Princess Sumaya University for Technology |
| PWM | Pulse-width modulation |
| SCORM | Shareable Content Object Reference Model |
| SPDT | Single-pole double-throw |
| STEM | Science, technology, engineering, and mathematics |
| UHU | University of Huelva |
| UI | User interface |
| UNED | Spanish National University for Distance Education |
| USB | Universal Serial Bus |
| USGM | Università degli Studi Guglielmo Marconi |
| VCC | Vapor compression cycle |
| VDC | Voltage direct current |

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
