# Peer review of "The Challenge of Digital Transition in Engineering. A Solution Made from a European Collaborative Network of Remote Laboratories Based on Renewable Energies Technology"

_asi, doi:10.3390/asi6020052_

Round 1

Reviewer 1 Report

This paper describes the lab part of the online teaching of engineering courses by incorporating virtual and real labs through remote access to online courses. Therefore, the advantage of digital transformation of higher education by creating a standardised educational model based on remote renewable energy laboratories in EU universities.

The paper is well written and the figures are illustrative of the idea. However, it is important to state how many simultaneous labs could be conducted in virtual or real mode using remote access. This will show the limitation of this platform. Online lab is one of the long awaited domain during COVID-19 days. However, this technology is not yet mature.

Evalaution of the online labs is a BIG problem. We can not rely on student's surveys. You need to measure it with some standard scale and the yardstick must be comparable to onsite labs to judge the effectiveness of the psychomotor learning.

It will be interesting to know how the learning outcomes are evaluated for an online/virtual lab?

Is it possible to conduct open ended labs (OELs) using virtual mode?

Reviewer 2 Report

The article is devoted to describing experience of a multi-university project in creating remote labs for studying topics related to renewable energy sources. This is an important topic to teach, but the article has serious problems that should be fixed to make its contribution clear and the conclusions sound.

The major problems that need fixing are:

1. Creating virtual laboratories is an old and well-researched area. But the article has a rather small reference list (and some references lack information about venues of publishing). I strongly suggest expanding the review of related work and concentrate more on discussing what was unique in author's experience and where they confirmed (or contradicted) findings of other researchers. You may also want to begin a separate section "Related work" around line 87.

2. The authors regularly state that their chief difference is creating standardized infrastructure, however I don't find descriptions or references to the standards that were created during this study. This makes the study look more a cooperative effort of a few universities to create a shared educational resource than an attempt to develop industry-wide standards. Please, describe the standards you are creating (or have been created already) if you want to claim creating standardized products.

2.1. In particular, the authors state that "Based on students’ and professors´ answers  ... acceptance that they have with the creation of a standardised infrastructure of remote laboratories between different European universities (Q0; AVG = 4.8; AVG = 4.82)" - however Q0 doesn't ask anything about standardised infrastructure, it only asks "Being able to access other remote labs from other universities is useful" which doesn't require standardizing. If you want to make a conclusion about standardizing, you should let your study participants access remote labs from other universities via standardised and non-standardised infrastructure, and the ask which experience was better. 

2.2. Instead of focusing on standardisation, you can enhance the article by providing advanced discussion of the surveys results, e.g.,  why students rated the manual and the possibility of using a webcam significantly lower than teachers and what was done to identify and fix the problems with the manual, etc. Receiving more explanations about the questions which students and/or teachers answered particularly low can become a good scientific contribution of this article.

3. In lines 115 and later, the authors describe a reference [15] which seems to be about (the quality of English doesn't let me understand it fully) a set of virtual labs hosted by different universities (PSUT, MU, HU and JUST). But in lines 140-141 the authors state "Inside these few works, only one offers a remote laboratory network, integrated by four laboratories; the four laboratories are placed at the same university" which contradicts what was described in the reference [15]. This is important for justifying the soundness and novelty of the research, so it needs fixing and more explanation.

4. Figure 3 and the text around it seem to present some of the results that were obtained by Munster Technological University in a joint project, however there are no authors from that university in this paper and there's no citing of sources either. Please, clarify, why do you publish results obtained by surveys from another university without citing references. In particular, who comprised the "consortium" that distributed surveys presented there and was their contribution to the article reflected in the authorship?

5. Reading the description of your remote lab, I don't see much controls for the student (besides selecting the load for discharging the capacitor), so the number of possible experiments is very limited. Why did you decide to give the students remote access to physical equipment then if you could just record all the possible experiments once and show their results to the students? This will use the equipment less and allow any number of simultaneous "lab experiments". Giving students direct access to manipulating your remote laboratory makes sense when they can create a lot of setups so that you cannot prepare the data for each possible variant in advance.  But this is not the case. Please, describe, why did you make this choice and do you see it as justified.

The article is written very poorly; it requires thorough English editing - in some places, I could not understand what the authors meant or the text seemed to contradict their intentions. E.g., in line 148 you write "each laboratory is placed at each university" - did you really mean it or did you mean that each laboratory is placed at only one university? Other examples of problematic English are: "In a try to join education and inclusion", "This laboratory is physically located at PSUT; a fuel cell training plant, where students learn concepts...", “Self – Check Theorical part” (a typo?),  "Q6 How can you the graphs?", "with online teaching, i.e. hybrid education, taking centre stage". Please, edit the article thoroughly for clarity.

There are small technical problems in the article:

1. Figure 1 has four parts that are described in the nearby list - please mark parts of Figure 1 (a), (b)...(d) and mention them in the relevant section.

2. Figure 1 and the list before it describe 4 laboratory types; however laboratory types in Table 1 are different. Please, use the laboratory types from Figure 1 in Table 1.

3. lines 175-176  - "merged traditional and remote learning" in which way?

4. Fig 2 - font is too small

5. lines 208-209 mention using Moodle while in Table 1 authors' proposal doesn't mention it. Please make your proposal consistent.

6.   Fig 5  can be improved by making labels for connections, showing the data that is transmitted between the nodes.

7. line 291 - Why did you choose a relatively old SCORM standard instead of more modern IMS LTI? Please explain.

8.  Figures 9,10,11,12 are excessive - there is no need to provide trivial screenshots of logging in to a system in a scientific article; just listing standard Moodle features you used (e.g., external database authorization) is enough

9. Abbreviation AR is in the list but I didn't find it in the article text

The aim of this work is important, but it requires serious revision in order to become a valuable and impactful contribution to scientific knowledge.

Round 2

Reviewer 1 Report

Author's have significantly improved after incorporating all necessary ammendments. Congratulations on such a good work.

Reviewer 2 Report

The article was significantly improved, but some of the problems noted in the previous review persist. In particular

Response 2.2.1. If you write "it is not the objective of the proposal to study and compare a standardised and no standardised infrastructure" then you shouldn't make conclusions like (line 425) "it can be seen the acceptance that they have with the creation of a standardised infrastructure of remote laboratories between different European universities". Consider rewording this conclusion to match the question your asked (e.g., "it can be seen that students and professors consider being able to access remote labs from other universities useful". This is important for scientific soundness.

Response 2.5 You should discuss and address the problem that there's no visible difference between virtual and real remote laboratories. If you believe that "In engineering careers, the goal is not to minimize or prevent the use of the laboratory, but quite the opposite. " then this certainly needs been discussed. If your goal is to let students practice with physical equipment (which is important), you need on-site laboratory - remote laboratory cannot let them perform physical manipulation with equipment. And if you use a remote laboratory, virtual laboratory becomes indistinguishable from a physical one - the experience of students is the same. This can have negative consequences for engineering education (depending on what you want for your students: learn to work with equipment or just see experimental data supporting a theory they learn) and it can limit using remote laboratories to certain kind of courses (e.g., they cannot be used in courses teaching the equipment that graduates will be expected to handle on-site during work). Discussing the limits of applicability of your method will improve the article.

Response 2.6 Some English errors remained (e.g. "Q6 How can you [do what?? - reviewer] the graphs?" (table in line 414). There are also errors in the new text (e.g., "we can observe that students evaluate with a lower punctuation [??] than professors the question Q2"

Response 2.7.7  If some of the decisions described in the article like choosing SCORM packages were made by an external company, you should give credit to the company for those decisions in the article text. By default, it is assumed that the decisions described in the article are made by the authors.
